# Canola Seed Protein: Pretreatment, Extraction, Structure, Physicochemical and Functional Characteristics

**DOI:** 10.3390/foods13091357

**Published:** 2024-04-28

**Authors:** Huipeng Zhu, Lu Wang, Xiaoyu Li, John Shi, Martin Scanlon, Sophia Xue, Matthew Nosworthy, Nazanin Vafaei

**Affiliations:** 1Nano-Biotechnology Key Laboratory of Hebei Province, Skate Key Laboratory of Metastable Materials Science and Technology, School of Environmental and Chemical Engineering, Yanshan University, Qinhuangdao 066004, Chinawl@ysu.edu.cn (L.W.); 2Guelph Research and Development Center, Agriculture and Agri-Food Canada, Guelph, ON N1G 5C9, Canada; jun.xue@agr.gc.ca (S.X.);; 3Faculty of Agricultural and Food Science, University of Manitoba, Winnipeg, MB R3T 2N2, Canada

**Keywords:** canola seed protein, essential amino acids, pretreatment, extraction, physicochemical properties

## Abstract

The rapid growth of the global population has led to an unprecedented demand for dietary protein. Canola seeds, being a widely utilized oil resource, generate substantial meal by-products following oil extraction. Fortunately, canola meals are rich in protein. In this present review, foremost attention is directed towards summarizing the characteristics of canola seed and canola seed protein. Afterwards, points of discussion related to pretreatment include an introduction to pulsed electric field treatment (PEF), microwave treatment (MC), and ultrasound treatment (UL). Then, the extraction method is illustrated, including alkaline extraction, isoelectric precipitation, acid precipitation, micellization (salt extraction), and dry fractionation and tribo-electrostatic separation. Finally, the structural complexity, physicochemical properties, and functional capabilities of rapeseed seeds, as well as the profound impact of various applications of rapeseed proteins, are elaborated. Through a narrative review of recent research findings, this paper aims to enhance a comprehensive understanding of the potential of canola seed protein as a valuable nutritional supplement, highlighting the pivotal role played by various extraction methods. Additionally, it sheds light on the broad spectrum of applications where canola protein demonstrates its versatility and indispensability as a resource.

## 1. Introduction

An increasing global population combined with nutritional demand for protein has resulted in an upward trend year over year. It is estimated that the investment of protein globally has increased from USD 25.65 to USD 48.77 billion between 2016 and 2025 [1]. Animal-based sources have been recognized as high in protein content, capable of meeting or exceeding the protein and amino acid requirements of humans. Nevertheless, animal protein has some inherent limitations on its broad application such as high economic cost and voluntary dietary restrictions (vegetarians, vegans, etc.) [2]. More importantly, some persons are allergic to animal products leading to an inability to consume these protein sources [3,4]. Additionally, long-term intake of animal protein has been demonstrated to increase the risk of non-communicable chronic diseases such as cardiovascular disease and type 2 diabetes [5,6]. One of the reasons for the popularity of plant-based protein is that consumers are becoming more concerned about consuming a healthy diet as well as environmental sustainability [7]. Similarly, a concomitant rise in vegetarianism and new research findings in nutrition focusing on the beneficial attributes of plant-based protein as related to biodegradability, sustainability, environmental claims, etc., has resulted in an increasing push for human consumption of plant-based protein sources rather than animal-based ones [8]. These factors have resulted in a rapid increase in consumers’ preference for plant-based protein as an alternative to animal-based protein [9]. Plant-based proteins are also being investigated as new candidates for synthetic biopolymers, which has generated significant interest from the global packaging industry in the last decade [10]. Importantly, the demand for plant-based protein is largely exceeding the current protein production.

Crops such as cereals, legumes, pulses, and nuts are becoming more recognized as high-quality sources of proteins, and, while they are extensively cultivated all over the world, they still cannot meet the global demand. Canola seeds (*Brassica napus*) are one of the most common plant-based oil sources worldwide, possessing the advantages of easy cultivation, rich nutritional qualities, and good environmental sustainability. Regretfully, the canola seed meal after oil extraction is considered a ‘waste product’ and is primarily discharged into the environment or used as feed for livestock animals. Notably, the canola seed meal is abundant in multiple biological substances, including protein, minerals, polyphenols, and cellulose. Currently, reviews in relation to canola seed for the most part focus on food applications, extraction, and function, while others summarize its nutritional value. With regard to structure and pretreatment, reviews are few or none. This review serves as a comprehensive synthesis of recent research, offering insights into the intricacies of pretreatment, extraction, structure, and physicochemical characteristics. By emphasizing the various applications of canola seed protein in food products, it contributes to the potential utilization of this natural resource for enhanced nutritional value and improved food product development.

## 2. Canola Seed

Canola, belonging to the Brassicaceae family, is one of the most widely cultivated crops due to its high yield with high nutritional value and high oil volume [11,12,13]. The demand for canola has expanded dramatically since the 1970s because of population growth, food consumption, and the demand for renewable energy [14]. Globally, the production of canola seed has risen from 29.7 million tons in 1994 to 72.3 million tons in 2020 (Figure 1). There are three main varieties of canola grown worldwide, including *Brassica rapa*, *Brassica napus*, and *Brassica juncea*. Canola seed is a mustard seed and is extremely abundant in oil and low in glucosinolates and erucic acid. As one of the most important oil crops in the world, canola seed accounts for 16 percent of vegetable oil production, ranking third only behind soybean oil and palm oil (Figure 2) [15,16]. Many dietitians believe that canola oil could be considered as the healthiest edible oil based on its unique characteristics such as fatty acid composition and levels of tocopherols, phytosterols, and polyphenols. Canola seed oil contains plenty of unsaturated fatty acids including linoleic (omega-6) and α-linoleic acid (omega-3) with a ratio of about 2:1, and it also contains a low amount of saturated fatty acids (<7%) compared to other common vegetable oils [17]. Therefore, canola seed oil is renowned for its unparalleled nutritional profile, positioning it as one of the most valuable edible oils. Beyond their richness in oil content, canola seeds boast a significant presence of essential amino acids, rendering them an exemplary dietary supplement [18,19]. Furthermore, the versatility of canola seeds extends to their applicability in various industrial domains, including biofuels, cosmetics, and other related products [20].

## 3. Canola Seed Protein

Beyond its oil content, canola seed contains a considerable amount of protein. After oil extraction, the resulting meal consists of 36–40% protein on a dry weight basis. Canola seed protein is mainly distributed in the seed embryo, which consists of two cotyledons. Canola seed protein can be divided into three parts (Figure 3), including the inactive storage protein, membrane protein, and enzyme-active substances [21]. The inactive storage protein in canola seed is composed of cruciferin (12S), napin (2S), and oleosin, which account for 60%, 20%, and 8%, of the total protein content, respectively [22,23]. Cruciferin is a neutral protease with a molecular weight of 300–310 kDa and an isoelectric point of 7.2. It is made up of six subunits, and each of them has a molecular weight of approximately 50 kDa [24,25]. Each submit of cruciferin contains two polypeptide chains: a 30 kDa acidic chain with 254–296 amino acid residues and a 20 kDa basic chain with 189–191 amino acid residues, with the two polypeptide chains being connected by a single disulfide bond [26,27]. Napin is a fundamental protein characterized by a molecular weight distribution spanning 12.5–14.5 kDa, accompanied by an isoelectric point of approximately 11.0. The structural composition of napin reveals an elevated concentration of amidated amino acids [28]. It is composed of two polypeptide chains; one comprises about 40 amino acid residues with a molecular weight of 4.5 kDa, and the other comprises about 90 amino acid residues with a molecular weight of 9.5 kDa. Both the inter and intra chains in napin are stabilized by disulfide bonds [29]. Oleosin is a predominant lipid transfer protein, specifically reigning as the principal constituent within the oil body protein profile [24].

The secondary structure of canola seed protein is more complicated than the primary structure. In the composition of canola seed protein, a substantial proportion is characterized by α-helix formations (~10%) and β-sheet configurations (~50%), accompanied by a minor presence of random coil and β-turn structures [30,31]. For cruciferin, the hydrophobic β-sheet conformation, constituting 50% of the overall secondary structure, is positioned within the protein’s interior. Conversely, α-helix conformations are situated on the protein’s surface [32]. Different from cruciferin, napin contains about 40–46% α-helices and about 12% β-sheets [33]. In general, the high proportion of β-sheet in the napin fraction results in low accessibility to the digestive enzymes during gastrointestinal digestion, thereby leading to reduced bioavailability. Therefore, the ratio of α-helix to β-sheet is significant in influencing the nutritive value of canola seed protein [34]. Typically, the assessment of nutritional protein quality involves a comparison between total protein content and amino acid composition with established standard reference values [35]. Nevertheless, the bioavailability of dietary proteins is intricately tied to their hydrolysis by digestive enzymes in the gastrointestinal tract, a process markedly influenced by the inherent protein structure, including the secondary structure and nutrient matrix. Therefore, the scrutiny of protein secondary structure holds utmost importance. 

Protein secondary structure can be characterized using various methods [36]. The first one used is Fourier-transform infrared spectroscopy (FT-IR). However, this method may introduce inaccuracies in protein secondary structure analyses, as it solely provides structural features of chemical groups within the spectrum [32]. The second method approach utilizes standard sphere-source FT-IR spectroscopy for scrutinizing protein secondary structure. Despite its utility, this method has limitations in elucidating the chemical properties of microbial materials measuring < 20–100 microns (depending on the type of infrared micro-spectrum). Additionally, the sphere-source FT-IR spectrum is susceptible to interference from other biological components, such as carbohydrates with scattering effects [37]. The third method involves estimating protein secondary structure using FT-IR micro-spectra with a synchrotron light source. However, it is important to note that this method, like the other two methods described, is not entirely free from inaccuracies. Theodoridou et al. [38] conducted an analysis of the structural characteristics of black and yellow canola seed proteins using synchrotron-based Fourier transform infrared micro-spectroscopy. The detailed findings from their study are presented in Table 1. The ratio of amide I to amide II can show the differences in the molecular structure of proteins from different sources. The protein from black canola seeds had higher amide I and amide II area values compared to that from yellow canola seeds, while no differences were observed in the height of amide I to amide II. In terms of the content of β-sheets, the highest level was found in the protein from black canola seeds (Table 2). The relative composition of the secondary structure in a protein, such as α-helices and β-sheets, significantly influences the nutritional value, quality, and digestive ability of the protein. For instance, a high percentage of β-sheets in the secondary structure may partly lead to lower access for gastrointestinal digestive enzymes, thus lowering protein bioavailability. If the ratios of α-helix to β-sheet in the intrinsic secondary structures between the proteins differ, their nutritional values may therefore also differ [39]. 

Canola seed protein is rich in glutamic acid, aspartic acid, and leucine [40,41]. Though the composition and content of amino acids in canola seed protein are similar to those in soybean protein, the protein efficiency ratio (PER), an indication of the efficiency of a protein source for mammalian growth, of canola seed protein, 2.64, was higher than the one of soybean protein with the value of 2.19 [42]. Based on the PER value alone, canola seed protein can be considered beneficial for human consumption [43,44]. Nevertheless, the existence of glucuronic acid, phytic acid, and other anti-nutritional factors can reduce the digestibility of canola seed protein in the gastrointestinal tract, which can subsequently impact the utilization of canola seed protein [45,46]. 

More efforts have been dedicated to addressing the limitations associated with canola seed protein. Firstly, novel canola varieties have been developed using modern breeding technology, such as the international backcrossing program, to reduce the levels of glucuronic acid in canola seeds [47,48]. Secondly, nitrogen fertilizer often induces the accumulation and concentration of protein in canola seeds [49]. Thirdly, the application of pretreatments and fungal fermentation post-harvest can enhance the nutritional quality of canola seed protein [50]. Furthermore, the environment in which canola is cultivated can significantly impact its protein fraction. Gunasekera et al. [51] investigated the effects of environmental factors on the protein content of canola seeds under field conditions in the Mediterranean region and the southwest of Australia. Their findings indicate that both high temperature and drought can increase the concentration of canola seed protein.

## 4. Pretreatment and Its Impact

Canola seed protein, derived as a by-product from canola seeds (canola meal) post oil extraction, exhibits commendable nutritional attributes. In the area of natural product processing, the optimization of extraction efficiency and duration for the protein fraction necessitates the application of effective pretreatment methodologies. Indeed, several pretreatment techniques have been systematically employed to promote both the extraction yield and physicochemical characteristics of canola seed protein. In this context, the following three pretreatment techniques will be discussed: pulsed electric field, microwave irradiation, and ultrasound.

### 4.1. Pulsed Electric Field (PEF)

As an energy-efficient and economically viable non-thermal technology, pulsed electric field (PEF) treatment has garnered significant attention in both comprehensive studies and practical applications within the broader food processing industry [52,53]^.^ This sophisticated procedure encompasses the application of a high-intensity electric field to the target sample, which is positioned between two electrodes. The electric field is delivered in the form of precisely controlled pulses, with each sample being treated over an exceptionally brief duration [54]. PEF is performed by the combination of a high electric field intensity (10–50 kV/cm), short pulse width (0–100 μs), and high pulse frequency (0–2000 Hz) to treat liquid and semi-solid materials, which favorably forms a production line of continuous sterilization and aseptic filling. When compared to traditional treatment technology, PEF possesses the advantages of environmental protection, low energy consumption, and adjustability depending on the sample being treated [55]. Zhang et al. [56] studied the effect of PEF pretreatment on the structure of canola protein. The results indicated that the parameters of PEF including voltage and treatment time had important impacts on the secondary structure of canola seed protein (Table 3). In the amide I region of canola protein, the higher proportions of α-helices and β-sheets and the lower proportions of β-turns and random coils showed that the secondary structure of canola protein was significantly influenced by PEF treatment [56]. The voltage increase could decrease the proportions of α-helices and β-turns and increase the proportion of random coils. This study demonstrated that PEF pretreatment significantly enhanced the functional properties of canola protein and its fractions, including its solubility, water-holding capacity, emulsifiability, emulsion stability, oil-holding capacity, foamability, and foam stability. Infrared spectrometry indicated alterations in the protein’s secondary structure post-PEF with shifts in the proportions of α-helices, β-sheets, and β-turns within the amide I region. However, it was observed that voltage alteration had a lesser impact on the amount of β-sheets. Consistent with voltage, treatment time followed similar patterns. In another study, it was observed that an increase in voltage and extension in treatment time led to a simultaneous decrease in the proportion of α-helices and an increase in the proportion of β-sheets within egg protein [57]. 

### 4.2. Microwave Treatment (MC) 

The term “microwave” pertains to electromagnetic waves with frequencies ranging from 300 MHz to 300 GHz. These waves find extensive applications in various sectors, including food industries, chemical industries, pharmaceutical industries, and more [58,59,60,61]. Microwave exposure can induce molecular vibrations through both long and short wavelengths and high frequencies, resulting in the fragmentation of plant cell walls [62]. Consequently, microwave exposure has been employed to enhance the separation of proteins through electromagnetic effects [63]. Li et al. [64] investigated the impact of microwave pretreatment on proteins isolated from canola meals following supercritical carbon dioxide extraction of oil. Surprisingly, the secondary structures of canola seed protein remained unchanged, including α-helices and β-sheets, after microwave pretreatment. However, there were variations in the amino acid compositions of canola seed protein, as detailed in Table 4. The significant thermal energy derived from the conversion of electromagnetic energy in microwaves has the capacity to induce the denaturation of specific proteins, leading to a subsequent reduction in the quantities of amino acids. In the context of canola seed protein, microwave treatment not only affects their structural integrity but also exerts a discernible influence on various physicochemical properties. These properties encompass solubility, foaming capability, water/oil holding capacity, emulsion surfactant capacity, and stability, among other factors [64].

### 4.3. Ultrasound Treatment (UL)

Ultrasound, classified as an environmentally friendly physical technology, constitutes an acoustic wave with a frequency exceeding 20 kHz [65]. The utilization of ultrasound technology has garnered increasing interest in the field of food industries [66]. Ultrasound induces cavitation, a dynamic process that disrupts cell walls and facilitates the release of trapped compounds from cells into the extraction medium, thereby significantly enhancing the extraction rate [67,68,69,70]. It has been reported that ultrasound induces the unfolding of protein structure that influences the function of a protein [71,72]. In the initial stage of their study, Li et al. [64] subjected canola seeds to ultrasound pretreatment, subsequently proceeding to isolate and characterize the proteins from the seeds post oil extraction. Similarly, Flores-Jiménez et al. employed ultrasound for the pretreatment of canola seeds. Following ultrasound pretreatment, a higher quantity of associated proteins was liberated, attributed to the disruption of chemical bonds. Consequently, the resulting protein fraction exhibited an elevated content of branched-chain amino acids, as outlined in Table 5 [73]. Essential amino acids characterized by more rigid structures exhibit close associations with other substances within plant tissues, resulting in higher contents in ultrasound-treated canola seed protein (UL-CSP) compared to untreated CSP [74]. Additionally, similar to microwave processing, ultrasound pretreatment did not alter the secondary structure of canola seed protein [64].

## 5. Extraction Method

The extraction and isolation of canola protein constitute a methodical procedure designed to procure purified protein fractions from canola seed meal. The significance of canola seed protein as a by-product emerges subsequent to the extraction of oil from canola seeds. The comprehensive process involves separating the protein-rich canola seed meal from the residual components following oil extraction. This by-product, abundant in protein, becomes the focus of extraction and isolation processes, yielding a valuable canola protein isolate with various applications in food and industrial applications.

Common canola protein extraction methods include aqueous extraction, where water is used as a solvent, or alkaline extraction, involving the use of alkaline solutions. These methods aim to solubilize proteins and separate them from other components in the canola meal. While the physicochemical properties of canola seeds have some similarities to those of soybeans, employing the conventional wet processing method applied to soybeans for canola seeds yields a lower output. This discrepancy is attributed to the broad isoelectric point range of canola seed protein, the presence of anti-nutritional compounds, and the harsh conditions encountered during the degreasing process of canola seeds [75]. The canola meal is usually processed by a pressing treatment to remove the residual oil. Compared with hot pressing (26%), cold pressing (temperature < 40 °C) has a higher protein recovery (45%). Östbring et al. [76] reported that protein isolated from cold-pressed materials had better emulsifying properties. Therefore, the meal derived from the cold-pressed canola seeds is used for protein extraction.

The extraction of protein from canola seeds typically involves several distinct categories, including alkaline extraction, isoelectric precipitation, acid precipitation, micellization (salt extraction), and dry fractionation and tribo-electrostatic separation. A commonly employed approach is the combination of alkaline extraction with isoelectric precipitation for the isolation of canola seed protein [77]. In this process, a solution of high alkalinity is utilized to dissolve the protein, followed by adjustment to the isoelectric point. Dissolving the protein in an alkaline environment induces strong repulsion due to the negative charge on the protein, resulting in its solubilization in the extraction solvent [78]. Alkaline extraction is frequently coupled with isoelectric precipitation to enhance protein extraction efficiency. Isoelectric precipitation, as an extraction method, operates by adjusting the pH of the protein solution to match its isoelectric point [79]. This adjustment renders the net charge of the protein nearly zero, minimizing its solubility and facilitating efficient protein separation [80].

Micellization serves as a crucial process involving the dissolution of protein in a nearly neutral salt solution, subsequently followed by recovery achieved through the reduction of ionic strength in the salt solution. This reduction can be accomplished through membrane separation or by diluting the precipitate at a lower temperature [81]. The employment of a salt solution as the extraction solvent proves instrumental, as it enables the complete dissolution of the protein, thereby elevating the overall extraction efficiency with a distinctive characteristic of micellization precipitation [82]. The utilization of a salt solution not only facilitates the full dissolution of the protein but also contributes to an enhanced extraction capability, further emphasizing the unique and advantageous features associated with micellization precipitation in protein extraction processes.

Dry fractionation and tribo-electrostatic separation are physical methods and can also be used to extract canola seed protein. Dry-fractionation technology, as a sustainable process, has been well developed due to its various advantages such as excluding water, excluding chemicals, no drying measures required, and low consumption [83]. Not only that, but the method can also maintain the natural characteristics of proteins while maintaining low energy and water consumption [84]. Yet, due to the fact that dry fractionation only allows a relatively small increase in protein content but is effective in removing fiber content, it is clearly more suitable for reducing fiber content rather than increasing protein content [85]. The obstruction of strong electrostatic interactions can also affect the separation of dry fractionation. Therefore, some researchers combined dry fractionation with another physical method called tribo-electrostatic separation to extract canola seed protein [86]. As a technique for separating particles based on the size and type of particle charge, tribo-electrostatic separation isolates finely ground plant materials into parts rich in protein, starch, or fiber, and has the same advantages as dry fractionation [87]. By adjusting parameters such as particle collision frequency, wall material, and charge mass of particle charging time, the protein content in the tribo-electrostatic separation can be maximized [88].

## 6. Techno-Functional Characteristics

Canola seed protein stands out as a nutritionally advantageous component, offering a notable array of benefits. Abundant in essential amino acids, with a particular emphasis on lysine, and featuring a well-balanced amino acid profile, it plays a pivotal role in facilitating protein synthesis and maintenance within the body. Beyond its nutritional prowess, canola seed protein serves as a valuable source of plant-based protein, rendering it an excellent choice for individuals adhering to vegetarian or vegan diets.

The solubility characteristics of canola seed protein add to its versatility in various applications within the food industry. This solubility is intricately influenced by factors such as pH, temperature, and ionic strength. In general, canola protein isolates exhibit commendable solubility under neutral to slightly alkaline conditions, broadening their applicability across a diverse range of food formulations.

### 6.1. Solubility

Protein solubility is related to the interactions between protein and solvent, such as the hydrophobic effect, electrostatic interactions, and hydrogen bonding [89,90]. In general, proteins are hydrophilic, and their solubility is always influenced by the pH of the solvent, especially in terms of their isoelectric point. Unlike soybean protein, the solubility of canola seed protein changes little in its isoelectric point range; only 40~50% of canola seed protein can be precipitated at its isoelectric point. In the nitrogen solubility curve of canola seed protein, there are two lowest solubility points at pH 4 and pH 7, which is an important physicochemical feature of canola seed protein that is different from other plant proteins. At higher pH, proteins with more net negative charges would contribute to the dissociation of protein aggregates, leading to improved solubility of the protein. Similarly, at lower pH, the increased net positive charge also facilitates the solubility of proteins [91]. Extensive studies have found that different pretreatment techniques including PEF, microwave, and ultrasound treatments would induce cell disruption and thus improve the solubility of the protein, consequently promoting an increase in protein yield and access to proteins with low molecular weights [64].

Following PEF treatment, the solubility of canola seed protein exhibits an increase from 43.25 to 50.07%, correlating with the rise in voltage (as illustrated in Figure 4). However, surpassing a residence time of 180 s leads to a subsequent decrease in solubility. Microwave treatment serves to augment the solubility of canola seed protein significantly, elevating it from 18.73 ± 1.83% to 36.70 ± 1.98% (as detailed in Table 6) [64]. Notably, the microwave-treated canola seed protein demonstrates a heightened concentration of histidine residues compared to non-pretreated counterparts, suggesting a potential association with the observed increase in solubility.

In contrast, ultrasound treatment induces the production of proteins characterized by a greater abundance of branched chains, resulting in an augmented branching degree [89]. The ultrasonic effect, leading to partial unfolding of protein molecules, is identified as the underlying mechanism responsible for the observed increase in protein solubility [92,93].

Furthermore, the study by Flores-Jiménez et al. [73] reveals variations in the solubility of canola seed protein isolates under different pH conditions and ultrasound exposure times (as depicted in Figure 5). It can clearly be seen from Figure 5 that it is a typical bell-shaped curve, with the lowest solubility at pH 4. And it can be found that, compared with the control treatment, the protein solubility significantly increased in the pH range of 6–8 for 30 min of ultrasound. This underscores the multifaceted influence of processing parameters on the solubility dynamics of canola seed protein.

### 6.2. Water/Oil Holding Capacity

Water-holding capacity (WHC) stands out as a paramount physicochemical attribute of proteins, defining their ability to retain water. This property plays a pivotal role in shaping the softness, tenderness, and other textural characteristics of protein-incorporated food products. Consequently, the interaction between proteins and water molecules significantly influences the flavor and texture of food items [94].

Several factors, including amino acid composition, molecular weight distribution, and advanced structural conformation, collectively contribute to determining the WHC of proteins. In general, proteins enriched with hydrophilic groups exhibit stronger WHC, as these groups can establish a greater number of hydrogen bonds with water molecules [95]. This interplay between protein characteristics and water-binding capabilities underscores the intricate nature of WHC.

Another critical property of proteins is their oil-holding capacity (OHC), a factor that profoundly influences the quality of food products containing proteins. OHC denotes the protein’s ability to retain oil, directly impacting its emulsification capacity. With an increase in protein concentration within a certain range, the hydrophobic groups of the protein intensify, resulting in enhanced OHC. This phenomenon occurs as the non-polar side chains of proteins intricately bind with the hydrocarbon chains of aliphatic compounds. The maintenance of food flavor, in general, is intricately linked to the OHC, emphasizing its significance in preserving the sensory aspects of food products [96].

Li et al. [64] found that canola seed protein possessed both good water and oil retention properties, with a water-holding rate of 293.3% and oil-holding capacity rate of 366.7%. Because of its excellent water and oil retention, canola seed protein can be used in meat products to reduce the overflow of water and fat during processing and ensure their taste and quality. Nevertheless, after pretreatment, the water/oil retention properties of canola seed protein are changed. Compared with the control untreated group, the WHC of canola seed protein was stronger at lower PEF parameter levels and shorter residence times, while the WHC of PEF-treated canola seed protein was weaker at higher levels of these parameters and longer residence times (Figure 4) [56]. However, in contrast to the WHC of treated canola seed protein, the OHC of canola seed protein was increased significantly with increasing pulse frequency and residence time of PEF treatment [56]. When the pulse frequency and residence time, respectively, increase to 800 Hz and 150 s, the OHC remains stable. Although canola seed protein possesses excellent water/oil retention capacities, the microwave and ultrasound pretreatments further improved on the water/oil retention performances significantly compared with non-pretreatment control samples (*p* < 0.05) as shown in Table 7. The increase in WHC is related to microwave and ultrasound pretreatments increasing the exposure of hydrophilic amino acids, expanding the structural conformation, and promoting the interaction between protein and water molecules, thus improving the water retention capacity of protein [97]. Similar to the WHC, the good OHC of the MV-treated and UL-treated canola seed protein may be attributed to those treated proteins having more hydrophobic and non-polar side chains, which can interact with more oil molecules. Both microwave and ultrasound pretreatments can induce the production of canola seed protein with side chains of non-polar residues that will interact with the hydrocarbon chains in fat molecules [98].

### 6.3. Emulsifying Properties

Proteins exhibit amphiphilic characteristics, with hydrophilic residues predominantly concentrated on the surface and hydrophobic residues primarily embedded within the protein structure. This amphiphilicity imparts proteins with notable interfacial activity. The evaluation of emulsification performance and characteristics of proteins often relies on two crucial indicators: emulsification activity and emulsion stability [99].

Emulsification activity and stability serve as key metrics to assess the protein’s ability to stabilize oil–water interfaces, quantified as the interfacial area stabilized per unit weight of protein. Canola seed protein, displaying an outstanding emulsification ability approaching 100%, emerges as a natural emulsifier suitable for preparing oil/water emulsions and enhancing emulsion stability. In a study conducted by Alashi et al. [100], the oil/water emulsion stabilized by a canola seed protein isolate demonstrated impeccable stability throughout storage.

The remarkable emulsification properties of canola seed protein can be attributed to various factors, including its solubility, hydrophobicity, and structural characteristics. In particular, the solubility of the protein exhibits a positive correlation with its emulsification properties. This interplay between protein solubility and emulsification underscores the multifaceted nature of protein functionality in emulsion systems.

In a study conducted by Wang et al. [101], it was observed that higher protein solubility contributes to a greater distribution of protein molecules at the oil–water interface, forming a thicker interfacial layer and thereby enhancing emulsification. Consequently, elevating protein solubility and regulating amphiphilicity emerges as a strategy to enhance protein emulsification ability. The heightened hydrophobicity of canola seed protein enhances interactions between protein molecules at the oil–water interface, resulting in the formation of a robust and thick interfacial layer that stabilizes the emulsion through spatial repulsion [77]. Canola seed protein, with its surface-rich hydrophobic groups, strengthens the binding capacity of the protein with oil droplets in emulsions, facilitating enhanced protein adsorption at the oil–water interface and, consequently, a more stable interfacial layer [102].

Beyond solubility and hydrophobicity, the inherent structure of the protein plays a pivotal role in influencing its interface properties. For instance, a higher sulfhydryl content and a greater proportion of β-sheet structures improve protein flexibility, facilitating stronger protein/lipid interactions and promoting protein rearrangement at the oil–water interface [103]. Similarly, a lower proportion of α-helix structures enhances protein flexibility, contributing to favorable emulsification properties [104]. Additionally, partial denaturation and the formation of disordered structures can further enhance protein adsorption at the oil–water interface [105]. The turbulent behavior and integration of oil vesicles create a more favorable orientation for protein distribution at the oil–water interface.

Recent studies indicate that the emulsification ability of canola seed protein is profoundly influenced by pretreatment. After PEF, microwave, and ultrasound pretreatments, both the emulsifying property (EC) and emulsifying stability (ES) of canola seed protein exhibit significant improvement, as illustrated in Table 7.

### 6.4. Foaming Properties

The phenomenon of foaming occurs at the interface between liquid and air, with a close relationship with the interfacial tension that exists between these two phases. The stability of foams is significantly influenced by key factors such as drainage, coalescence, and mismatch (coarsening). Furthermore, the protein’s capability to reduce interface tension and adsorb at the air–water interface assumes a critical role in the formation of stable foams [106].

The unfolding of the protein’s structure holds sway over its diffusion capacity at the air–water interface, thereby decisively shaping its foaming performance. Foaming stability is intricately connected to the presence of a mucilage layer that envelops bubbles [107]. The establishment of a resilient foam structural network is influenced by the water solubility of protein, an aptitude to readily concentrate at the liquid/gas interface, and the ability to form an adhesive layer characterized by sufficient viscosity and strength [108]. These attributes collectively contribute to the successful formation and maintenance of stable foams.

Canola seed protein, distinguished by its favorable physicochemical and structural attributes, exhibits outstanding foaming ability, reaching an optimal value of 102%. This achievement surpasses the foaming capability of soybean protein by a substantial margin [109]. Notably, the foams generated from canola seed protein maintain commendable stability even after extended storage periods. The exceptional foaming attributes of canola seed protein can be attributed to a combination of its physicochemical properties, primary and advanced structural features, and intricate electric interactions.

Research findings explored the influential role of pretreatments, such as pulsed electric filed (PEF) and ultrasound, in modulating the foaming capacity of canola seed protein. Significant variations in foaming performance were observed under different voltage and residence time conditions of PEF. In a study by Flores-Jiménez et al. [73], the pronounced impact of pH on the foaming capacity and stability of canola seed protein during various ultrasound exposure times was highlighted, as illustrated in Table 7. Ultrasound, through the induction of partial protein denaturation and the promotion of a more flexible structure in aqueous solutions, enhances the interaction between air and water interfaces, thereby amplifying the foam properties.

## 7. Conclusions

The global concern surrounding the scarcity of protein has intensified, prompting an urgent need for solutions. The large-scale cultivation of canola, while extracting oil, leaves behind a substantial amount of meal—a by-product abundantly rich in protein.

Through thorough investigations into canola seed protein, there is a noticeable trend towards utilizing post-oil extraction meals to produce canola seed protein, thereby elevating this economical source to value-added canola products. This transition not only enhances feasibility but also expands the applications of canola seed protein across various industrial sectors. This comprehensive review delves into the intricacies of the structure, pretreatment, extraction, and functional attributes of canola seed protein. However, future research endeavors should aim to unravel the biological and pharmacological activities inherent in canola seed protein. This deeper understanding will undoubtedly contribute to positioning canola seed protein as a valuable dietary supplement within the realm of functional foods.

## Figures and Tables

**Figure 1 foods-13-01357-f001:**
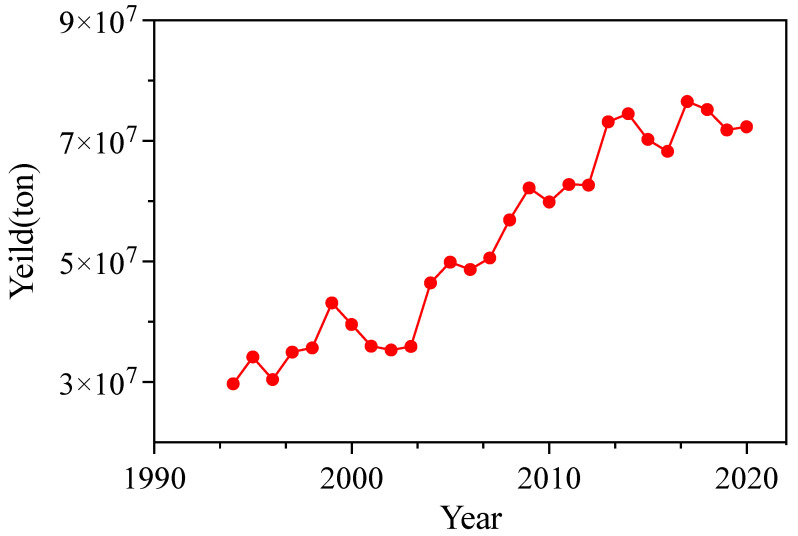
Annual production of canola seeds from 1994 to 2020.

**Figure 2 foods-13-01357-f002:**
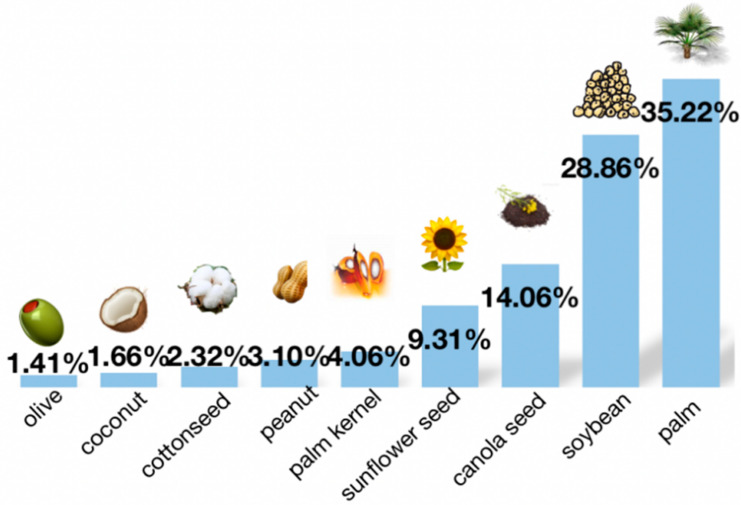
Global vegetable oil production structure in 2020.

**Figure 3 foods-13-01357-f003:**
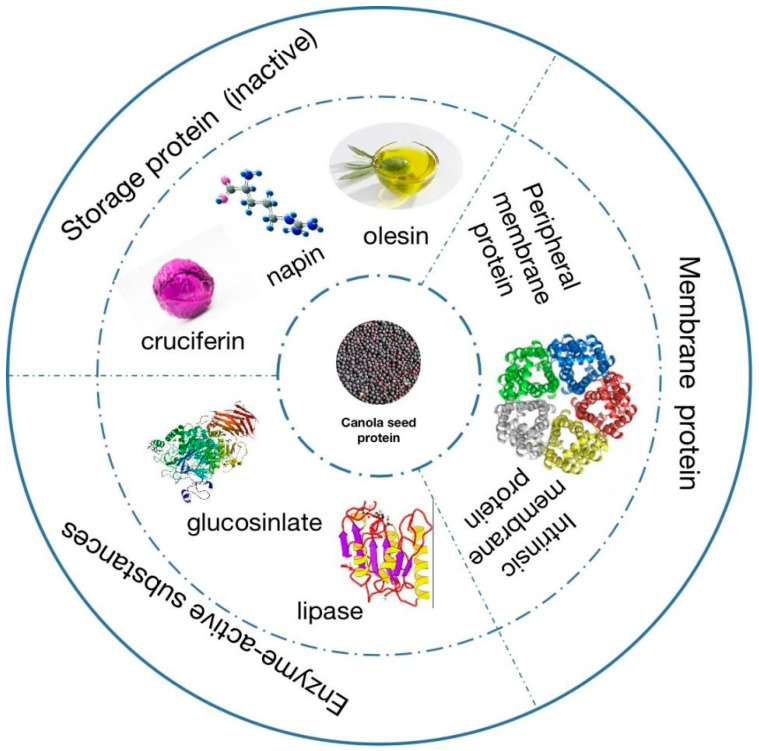
Composition of canola seed protein.

**Figure 4 foods-13-01357-f004:**
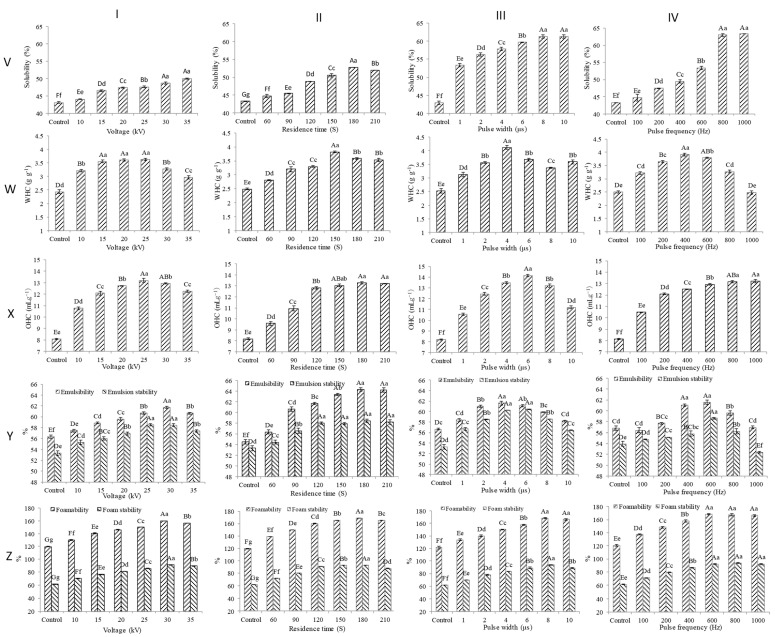
Effect of PEF pretreatment on functional properties of canola seed protein. The columns refer to PEF parameters: I, voltage; II, residence time; III, pulse width; and IV, pulse frequency. The rows refer to the functional properties of protein: V, solubility; W, water holding capacity (WHC); X, oil holding capacity (OHC); Y, emulsifiability and emulsion stability; and Z, foamability and foam stability [56]. Numbers bearing different letters in upper or lower case mean significant at *p* < 0.01 or *p* < 0.05, respectively.

**Figure 5 foods-13-01357-f005:**
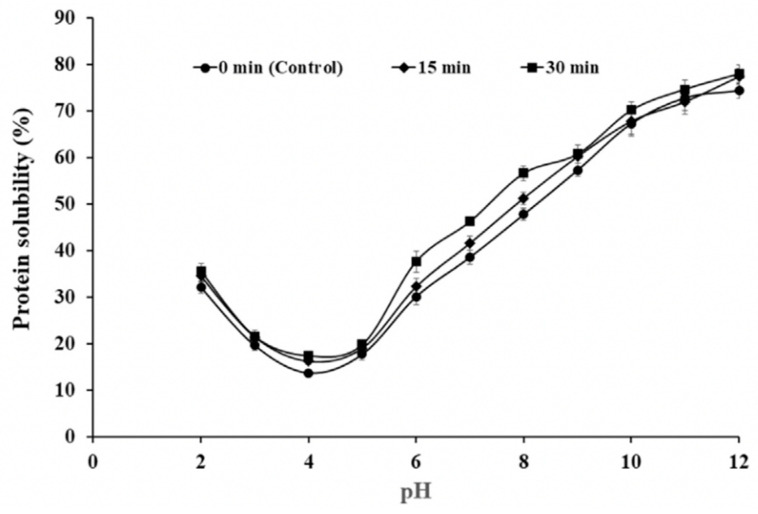
Effect of pH on the solubility of canola seed protein isolate at different ultrasound exposure times. Error bars show standard deviation [73].

**Table 1 foods-13-01357-t001:** The molecular structure spectrum profile of canola seed protein (Unit: Absorbance) [38].

Item	Yellow Canola Seed Protein	Black Canola Seed Protein
Amide I area	14.675 ^b^	17.775 ^a^
Amide II area	6.176 ^b^	7.549 ^a^
Ratio of amide I to amide II area	2.380 ^a^	2.356 ^a^
Amide I height	0.204 ^a^	0.247 ^a^
Amide II height	0.104 ^a^	0.125 ^a^
Ratio of amide I to amide II height	1.965 ^a^	1.989 ^a^

Means with different superscript lowercase letters within the same line are significantly different (*p* < 0.05).

**Table 2 foods-13-01357-t002:** Protein secondary structure profile [38].

Item	Yellow Canola Seed Protein	Black Canola Seed Protein
α-helix (height)	0.202 ^a^	0.246 ^a^
β-sheet (height)	0.170 ^b^	0.206 ^a^
Ratio of α-helix to β-sheet	1.184 ^a^	1.194 ^a^

Means with different superscript lowercase letters within the same line are significantly different (*p* < 0.05).

**Table 3 foods-13-01357-t003:** Effect of voltage and residence time on secondary structure of canola protein (portion, %) [56].

PEF Parameter		α-Helix	β-Sheet	β-Turn	Random Coil
Voltage	Control/kV	26.52	41.10	6.92	25.45
	10	23.96	42.77	6.26	27.01
	15	24.19	43.50	6.56	25.75
	20	22.60	43.94	5.56	27.90
	25	22.78	43.53	5.60	28.10
	30	28.53	44.15	5.38	21.94
Residence time	Control/s	26.51	41.10	6.92	25.45
	60	26.61	43.05	4.18	26.16
	90	28.15	35.18	3.92	32.76
	120	23.96	42.77	6.26	27.01
	150	23.06	42.13	6.77	28.04
	180	23.11	43.10	6.04	27.75

PEF: pulsed electric field.

**Table 4 foods-13-01357-t004:** Amino acid composition of canola seed protein after oil extraction under microwave pretreatment [64].

Amino Acid (g/100 g Protein)	CSP	Non-Pretreatment CSP	MV-CSP
Essential amino acid			
His	1.11	1.07	1.12
Ile	1.87	1.65	1.32
Leu	3.29	2.91	2.46
Lys	1.35	1.94	1.36
Phe	1.87	1.62	1.25
Thr	1.55	1.73	1.43
Val	2.24	2.09	1.78
Non-Essential amino acid			
Ala	2.48	2.31	2.01
Asn	3.43	2.66	2.18
Gly	2.28	1.85	1.66
Glu	7.26	4.38	4.26
Arg	3.00	2.35	2.12
Pro	1.55	1.13	1.05
Ser	1.59	2.09	1.51
Tyr	1.09	1.39	1.05

CSP: canola seed protein; MV-CSP: microwave-pretreated canola seed protein.

**Table 5 foods-13-01357-t005:** Amino acid composition of canola seed protein after oil extraction under ultrasound pretreatment [64].

Amino Acid (g/100 g Protein)	CSP	Non-Pretreatment CSP	UL-CSP
Essential amino acid			
His	1.11	1.07	1.33
Ile	1.87	1.65	1.82
Leu	3.29	2.91	3.23
Lys	1.35	1.94	2.03
Phe	1.87	1.62	1.71
Thr	1.55	1.73	1.92
Val	2.24	2.09	2.40
Non-Essential amino acid			
Ala	2.48	2.31	2.51
Asn	3.43	2.66	2.98
Gly	2.28	1.85	2.04
Glu	7.26	4.38	5.34
Arg	3.00	2.35	2.38
Pro	1.55	1.13	1.38
Ser	1.59	2.09	1.73
Tyr	1.09	1.39	1.47

CSP: canola seed protein; UL-CSP: ultrasound-pretreated canola seed protein.

**Table 6 foods-13-01357-t006:** Physicochemical parameters of CSP obtained from canola seeds after oil extraction with different pretreatments [64].

	CSP	Non-Pretreated CSP	MV-CSP	UL-CSP
Solubility (%)	16.04 ± 0.71 ^a^	18.73 ± 1.83 ^b^	36.70 ± 1.98 ^b^	23.63 ± 0.86 ^a^
Water holding capacity (%)	293.3 ± 15.3 ^a^	366.7 ± 30.6 ^ab^	416.7 ± 96.1 ^b^	416.7 ± 66.6 ^b^
Oil holding capacity (%)	366.7 ± 20.8 ^a^	263.3 ± 45.1 ^b^	456.7 ± 40.4 ^c^	466.7 ± 45.1 ^c^
Emulsion capacity (%)	100.02 ± 0.2	100.12 ± 0.22	100.00 ± 0.2	100.10 ± 0.19
Emulsion stability (%)	84.03 ± 0.50 ^a^	89.17 ± 1.04 ^b^	94.17 ± 1.04 ^c^	92.33 ± 2.0 ^c^

Means with different superscript lowercase letters within the same line are significantly different (*p* < 0.05). CSP: canola seed protein; MV-CSP: microwave-pretreated canola seed protein; UL-CSP: ultrasound-pretreated canola seed protein.

**Table 7 foods-13-01357-t007:** Effect of ultrasound exposure time at different pH values on the foaming capacity (FC) and foaming stability (FS) of canola protein isolates [73].

Ultrasound Exposure Time (min)	pH	FC (%)	FS (%)
0	2	220.1 ± 1.2 ^a^	68.1 ± 0.5 ^a^
15	2	220.17 ± 2.7 ^a^	68.7 ± 0.7 ^a^
30	2	219.7 ± 1.3 ^a^	67.9 ± 0.6 ^a^
0	4	173.1 ± 1.1 ^b^	84.8 ± 0.5 ^a^
15	4	175.4 ± 1.8 ^ab^	84.9 ± 0.1 ^a^
30	4	177.5 ± 1.2 ^a^	85.3 ± 0.1 ^a^
0	6	190.3 ± 2.7 ^b^	64.3 ± 0.7 ^b^
15	6	192.6 ± 0.8 ^b^	65.2 ± 0.6 ^b^
30	6	198.4 ± 1.4 ^a^	68.2 ± 0.6 ^a^
0	8	234.7 ± 2.4 ^b^	59.9 ± 0.5 ^b^
15	8	235.8 ± 2.5 ^b^	60.2 ± 0.6 ^a^
30	8	239.2 ± 1.1 ^a^	63.6 ± 0.4 ^a^
0	10	244.3 ± 0.9 ^a^	74.0 ± 0.5 ^a^
15	10	245.7 ± 1.1 ^a^	74.5 ± 0.5 ^a^
30	10	246.4 ± 1.3 ^a^	74.4 ± 0.7 ^a^

Means with different superscript lowercase letters within the same line are significantly different (*p* < 0.05). FC: foaming capacity; FS: foaming stability.

## Data Availability

No new data were created or analyzed in this study. Data sharing is not applicable to this article.

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
