# Peer review of "Canola Seed Protein: Pretreatment, Extraction, Structure, Physicochemical and Functional Characteristics"

_foods, 2024, doi:10.3390/foods13091357_

Round 1
Reviewer 1 Report
Comments and Suggestions for Authors
Zhu et al. aims to review the pretreatment, extraction, structure, physicochemical and functional characteristics of canola seed protein. Although the topic has been sufficiently reviewed, the following points need to addressed.
1. The title should be modified as “Canola seed protein: Pretreatment, extraction, structure, physicochemical and functional characteristics"
2. The abstract should be rewritten with elucidating point out of each subtopic discussed in this review, instead of a highlighting the general overview content of review.
3. In figure 1, “sustairable poor” should be replaced with “unsustainability” or “poorly sustainable”.
4. Figures 2 and 3 should combined as one figure with the latter preceded by the former as (A) and (B) parts of the same figure.
5. All the tables involving the data should be supplemented with copyright permission details from where they have reproduced from.
6. Table 6 should be properly formatted after removing the column head ‘canola meal’ (which can be mentioned in the title itself) by drawing a horizontal line to separate 4 different methods. Else it is not clear.
7. Figure 9 is too small and should be enlarged for clarity of labels.
8. All the references should be double-checked for adopting the journal style of referencing.
9. The authors should ensure with proving the full forms of all the abbreviations used in both tables and figures in their respective footnotes and figure captions.
10. Why only the Tables 1 and 2 are provided with statistical significance alphabets as superscript, while they are missing in Tables 3,4,5,7 and 8?
Comments on the Quality of English Language
Minor editing of English language required
Author Response
Manuscript ID: foods-2966924
Article Title: Canola seed protein: Pretreatment, extraction, structure, physicochemical and functional characteristics
Dear Editor and Reviewers,
We sincerely appreciate the feedback provided by the reviewers. We have diligently revised our manuscript in accordance with their recommendations and followed to the journal's requirements. Enclosed, please find a copy of the revised manuscript along with our responses to the reviewers' comments, detailing the revisions made point-by-point. We have made every effort to address all of the reviewers' concerns. The corrected and revised portions are highlighted in red.
The reviewers' comments were invaluable, and the resulting revisions have significantly enhanced the quality of the manuscript. We would like to extend our gratitude to the reviewers for their time and thoughtful remarks.
We are confident that the revised version of the manuscript fulfills the requirements of both the reviewers and the journal. We sincerely thank you for your time and for supporting our efforts to improve the manuscript.
Please feel free to reach out to us for any further clarification or adjustments needed.
We look forward to hearing from you.
Reviewers' comments:
- The title should be modified as “Canola seed protein: Pretreatment, extraction, structure, physicochemical and functional characteristics"
Response: As suggested, the title has been revised as “Canola seed protein: Pretreatment, extraction, structure, physicochemical and functional characteristics".
- The abstract should be rewritten with elucidating point out of each subtopic discussed in this review, instead of a highlighting the general overview content of review.
Response: The abstract section has been revised to clarify the key points of each subheading.
- In figure 1, “sustainable poor” should be replaced with “unsustainability” or “poorly sustainable”.
Response: The figure has been corrected as required.
- Figures 2 and 3 should combined as one figure with the latter preceded by the former as (A) and (B) parts of the same figure.
Response: We appreciate the suggestion. However, Figure 2 depicts vegetable oil production worldwide, while Figure 3 illustrates the composition of canola seed protein. Therefore, we believe it is more appropriate to keep them as separate figures.
- All the tables involving the data should be supplemented with copyright permission details from where they have reproduced from.
Response: Thank you for your useful advice. All tables have added references.
- Table 6 should be properly formatted after removing the column head ‘canola meal’ (which can be mentioned in the title itself) by drawing a horizontal line to separate 4 different methods. Else it is not clear.
Response: We have removed Table 6 and integrated its content directly into the manuscript
- Figure 9 is too small and should be enlarged for clarity of labels.
Response: Figure 9 has been modified with clarity labels according to reviewer’s suggestion.
- All the references should be double-checked for adopting the journal style of referencing.
Response: We have thoroughly reviewed and formatted all references in accordance with the journal's style guidelines.
- The authors should ensure with proving the full forms of all the abbreviations used in both tables and figures in their respective footnotes and figure captions.
Response: We have supplemented the full forms of the abbreviations as requested.
- Why only the Tables 1 and 2 are provided with statistical significance alphabets as superscript, while they are missing in Tables 3,4,5,7 and 8?
Response: We have added the statistical significance alphabet to Table 6 and 7. Regarding Tables 3, 4, and 5, the data does not demonstrate any statistically significant differences, rendering the inclusion of statistical significance alphabets unnecessary.
Reviewer 2 Report
Comments and Suggestions for Authors
Comments to authors:
The authors in this review presented the latest updated on canola protein characteristics and extraction methods. In fact, the review topic is very interesting and the authors presented the importance of extraction of canola seed protein as a valuable dietary supplement in a clear way.
The manuscript still needs to be revised and improved. A major revision is recommended.
Some comments are below:
- The authors better to provide the characteristics of the genetically modified of canola oil type.
- In Tables 1 and 2, there are some symbols without defining (a and b).
- Table 6 has a huge information in extraction method, I would suggest to be presented in a propriate format type. Also, some data has overlapped with other methods. They need to be divided depend on the conditions.
- The importance of study the effect of pH on protein solubility needs to be discussed with more explanation and Fig. 10 need to be cleared. The authors presented the effect of pH on solubility at different ultrasound exposure times but didn’t provide the importance of this study.
Author Response
Manuscript ID: foods-2966924
Article Title: Canola seed protein: Pretreatment, extraction, structure, physicochemical and functional characteristics
Dear Editor and Reviewers,
We sincerely appreciate the feedback provided by the reviewers. We have diligently revised our manuscript in accordance with their recommendations and followed to the journal's requirements. Enclosed, please find a copy of the revised manuscript along with our responses to the reviewers' comments, detailing the revisions made point-by-point. We have made every effort to address all of the reviewers' concerns. The corrected and revised portions are highlighted in red.
The reviewers' comments were invaluable, and the resulting revisions have significantly enhanced the quality of the manuscript. We would like to extend our gratitude to the reviewers for their time and thoughtful remarks.
We are confident that the revised version of the manuscript fulfills the requirements of both the reviewers and the journal. We sincerely thank you for your time and for supporting our efforts to improve the manuscript.
Please feel free to reach out to us for any further clarification or adjustments needed.
We look forward to hearing from you.
Reviewers' comments:
- In Tables 1 and 2, there are some symbols without defining (a and b).
Response: According to the reviewer’s suggestion, some symbols have been defined in Table 1 and 2.
- Table 6 has a huge information in extraction method, I would suggest to be presented in a propriate format type. Also, some data has overlapped with other methods. They need to be divided depend on the conditions.
Response:. We have removed Table 6 and integrated its content directly into the manuscript
- The importance of study the effect of pH on protein solubility needs to be discussed with more explanation and Fig. 10 need to be cleared. The authors presented the effect of pH on solubility at different ultrasound exposure times but didn’t provide the importance of this study.
Response: We have provided a more detailed explanation of the importance of studying the effect of pH on protein solubility. Additionally, we have clarified and analyzed Figure 10, elucidating the effect of pH on solubility at different ultrasound exposure times.
Reviewer 3 Report
Comments and Suggestions for Authors
The authors carry out a review of the structure, pretreatment methods and protein extraction of canola seeds due to their nutritional importance for society. It makes a contribution on the different extraction methods on the functional and structural properties of proteins, as well as their applications in industry for their use.
Greater clarity is required in the description of the extraction methods and in the discussion of their industrial applications.
1. Emails: Email addresses: lixiaoyu@ysu.edu.cn; wl@ysu.edu.cn and email addresses: john.shi@agr.gc.ca, Appropriately place the "*" or "**" symbol.
2. “Correspondence: author:” is incorrectly noted, “Corresponding author:” correct form
3. Eliminate “billion” from the first time, “between $25.65 billion and $48.77 billion between” as well as “millions” elsewhere in the manuscript.
4. In some lines the references are separated from the last word, correct according to the journal's citation criteria, and in some it includes the period “.” before the references and in others after.
5. Figure 4. The numerical values, in some they are very close or above the previous one with the next
6. In values like the following “12.5-14.5 kDa” change the middle dash to the following “-”
7. Figure 5 shows an image that has lost pixels, please improve it.
8. Figure 7. Homogenize the size and type of font throughout the manuscript. It looks very big. Review all the figures.
9. In table 6. It is not clear who is the referent of each text, to define well
10. In some lines the symbol “%” is repetitive as in the example “3.25% to 50.07%”, I suggest “3.25 to 50.07%
11. In Table 7, I suggest that “Percentage (%)” be placed in the first column and removed in front of “Solubility (%)”. . . And also. What are the units of the following four columns of the same table?
12. Correct: in “foaming capacity (FC)” to “Foaming capacity (FC)” and in “foaming stability (FS)” to “Foaming stability (FS)”
13. References 6 and 7 remove underline. Likewise 105 and 108
Author Response
Manuscript ID: foods-2966924
Article Title: Canola seed protein: Pretreatment, extraction, structure, physicochemical and functional characteristics
Dear Editor and Reviewers,
We sincerely appreciate the feedback provided by the reviewers. We have diligently revised our manuscript in accordance with their recommendations and followed to the journal's requirements. Enclosed, please find a copy of the revised manuscript along with our responses to the reviewers' comments, detailing the revisions made point-by-point. We have made every effort to address all of the reviewers' concerns. The corrected and revised portions are highlighted in red.
The reviewers' comments were invaluable, and the resulting revisions have significantly enhanced the quality of the manuscript. We would like to extend our gratitude to the reviewers for their time and thoughtful remarks.
We are confident that the revised version of the manuscript fulfills the requirements of both the reviewers and the journal. We sincerely thank you for your time and for supporting our efforts to improve the manuscript.
Please feel free to reach out to us for any further clarification or adjustments needed.
We look forward to hearing from you.
Reviewers' comments:
- Emails: Email addresses: lixiaoyu@ysu.edu.cn; wl@ysu.edu.cn and email addresses: john.shi@agr.gc.ca, Appropriately place the "*" or "**" symbol.
Response: We have made the necessary modifications to all email addresses as requested.
- “Correspondence: author:” is incorrectly noted, “Corresponding author:” correct form
Response We have corrected the it as suggested.
- Eliminate “billion” from the first time, “between $25.65 billion and $48.77 billion between” as well as “millions” elsewhere in the manuscript.
Response: The first "billion" has been deleted as suggested, see page 1.
- In some lines the references are separated from the last word, correct according to the journal's citation criteria, and in some it includes the period “.” before the references and in others after.
Response: All the references have been double-checked, and reformatted according to the journal requirements.
- Figure 4. The numerical values, in some they are very close or above the previous one with the next.
Response: The proximity or exceeding of numerical values in Figure 4 may be attributed to accidental errors.
- In values like the following “12.5-14.5 kDa” change the middle dash to the following “-”
Response:. “12.5-14.5 kDa” has been changed the middle dash to the following “-”.
- Figure 5 shows an image that has lost pixels, please improve it.
Response: The figure has been corrected as required.
- Figure 7. Homogenize the size and type of font throughout the manuscript. It looks very big. Review all the figures.
Response:. The figure has been corrected as required.
- In table 6. It is not clear who is the referent of each text, to define well
Response: We have removed Table 6 and integrated its content directly into the manuscript.
- In some lines the symbol “%” is repetitive as in the example “3.25% to 50.07%”, I suggest “3.25 to 50.07%”.
Response: “3.25% to 50.07%” has been changed to “3.25 to 50.07%”.
- In Table 7, I suggest that “Percentage (%)” be placed in the first column and removed in front of “Solubility (%)”. . . And also. What are the units of the following four columns of the same table?
Response: The table has been corrected as requested.
- Correct: in “foaming capacity (FC)” to “Foaming capacity (FC)” and in “foaming stability (FS)” to “Foaming stability (FS)”.
Response: As suggested, “foaming capacity (FC)” and “foaming stability (FS)” have been changed to “Foaming capacity (FC)” and “Foaming stability (FS)”.
- References 6 and 7 remove underline. Likewise 105 and 108.
Response: The underline has been removed as suggested.